# Application of LiDAR Derived Fuel Cells to Wildfire Modeling at Laboratory Scale

**Anthony A. Marcozzi** [1,*], **Jesse V. Johnson** [1], **Russell A. Parsons** [2], **Sarah J. Flanary** [2], **Carl A. Seielstad** [3] and **Jacob Z. Downs** [1]

1  Department of Computer Science, University of Montana, 32 Campus Drive, ISB 406, Missoula, MT 59812, USA; jesse.johnson@mso.umt.edu (J.V.J.); jacob.downs@umontana.edu (J.Z.D.)
2  US Forest Service, Rocky Mountain Research Station, Fire Sciences Laboratory, 5775 W. Highway 10, Missoula, MT 59801, USA; russell.a.parsons@usda.gov (R.A.P.); sarah.j.flanary@usda.gov (S.J.F.)
3  National Center for Landscape Fire Analysis, University of Montana, 32 Campus Drive, CHCB 428, Missoula, MT 59812, USA; carl@firecenter.umt.edu
*  Correspondence: anthony.marcozzi@umontana.edu

**Abstract:** Terrestrial LiDAR scans (TLS) offer a rich data source for high-fidelity vegetation characterization, addressing the limitations of traditional fuel sampling methods by capturing spatially explicit distributions that have a significant impact on fire behavior. However, large volumes of complex, high resolution data are difficult to use directly in wildland fire models. In this study, we introduce a novel method that employs a voxelization technique to convert high-resolution TLS data into fine-grained reference voxels, which are subsequently aggregated into lower-fidelity fuel cells for integration into physics-based fire models. This methodology aims to transform the complexity of TLS data into a format amenable for integration into wildland fire models, while retaining essential information about the spatial distribution of vegetation. We evaluate our approach by comparing a range of aggregate geometries in simulated burns to laboratory measurements. The results show insensitivity to fuel cell geometry at fine resolutions (2–8 cm), but we observe deviations in model behavior at the coarsest resolutions considered (16 cm). Our findings highlight the importance of capturing the fine scale spatial continuity present in heterogeneous tree canopies in order to accurately simulate fire behavior in coupled fire-atmosphere models. To the best of our knowledge, this is the first study to examine the use of TLS data to inform fuel inputs to a physics based model at a laboratory scale.

**Keywords:** LIDAR; TLS; fuel modeling; wildland fire modeling; coupled fire-atmosphere models

## 1. Introduction

Computational Fluid Dynamics (CFD) fire models are increasingly used to answer scientific questions related to fire behavior, weather effects, ecological impact, and firefighter safety [1,2]. While there is debate on the appropriate role of CFD models in scientific and operational modeling [3], there is also increasing interest in overcoming model limitations in order to apply coupled fire-atmosphere models to study and manage prescribed fires [4–6].

Prescribed fire is an important tool for mitigating the threat of wildfire to fire sensitive areas, such as the Wildland Urban Interface [7], as well as achieving landscape goals as a form of restoring natural disturbance regimes [8,9]. Physics-based, coupled fire-atmosphere models have been used to simulate grass fires [10,11], single tree burns [12], and more recently, complex prescribed fire and wildfire environments [6,13,14]. However, it is a challenge to describe three-dimensional fuel properties in coupled fire-atmosphere models. Fire modelers must balance trade offs between grid resolution, computational complexity, and data availability to achieve fidelity to limited observations.

This challenge is especially important in light of the important role fuel heterogeneity plays in fire behavior. Laboratory experiments have shown that gaps between fuel particles, and the gaps within an individual fuel particle, are important factors in determining

ignition when exposed to a heat source [15]. Additionally, the spatial distribution of bulk density within a tree canopy was found to significantly alter crown fire potential in a modeled environment [16]. Recent studies have demonstrated that modifying fuel distributions at the landscape level using fuel treatments leads to changes in simulated fire behavior [2,17,18]. One study found that increasing fuel fidelity and heterogeneity information impacted fine-scale wind discontinuities which reduced fire spread and area burned [19].

Understanding fuel heterogeneity through field sampling is important for determining fire behavior. However, fuel sampling has historically been concerned with capturing bulk fuel quantities to support steady-state rate of spread fire models such as the Rothermel model [20]. Fire behavior fuel models were classified through field sampling methods that captured weight per unit area of downed woody material, litter and duff, shrubs, and small conifers [21]. Current, operationally used, fire behavior fuel models capture broad fuel trends on a landscape, but overlook fuel heterogeneity in favor of simplifying assumptions that permit rapid calculations and data collection.

As our understanding of the importance of fuel heterogeneity has grown, there has been interest in developing methods which capture greater spatial detail, particularly with respect to variability in fuel structure and distribution. Recently, fuel sampling methods have been devised in order to capture 3D variability in fine scale fuels [22]. This work offers promising characterizations of fine-scale fuels which drive the behavior of low intensity surface fires often found in prescribed burn environments. However, these techniques are time consuming, labor intensive, and cannot be scaled to areas typical of prescribed fire units.

Another approach to obtaining high fidelity 3D models of vegetation are point clouds from Terrestrial Laser Scans. TLS has been used to map crown profiles [23], predict canopy fuel loading [24], and simulate 3D surface fuel beds [25]. In addition, point clouds from TLS are capable of generating precision tree models [26] which can be used with allometric equations for biomass estimates [27].

There are several challenges associated with correlating 3D point clouds to spatially explicit fuel attributes without the use of destructive sampling, and the relationship between point density and foliage mass, or bulk density, is an open research question. This relationship is difficult to quantify because point density is dependent on multiple factors related to the scanning target and the scanning environment. For example, point density is related to the scan angle, branch angle, occlusion, halo effect, and duplicate points from colocated scans [28,29]. A higher point cloud density does not necessarily correlate to a higher density of foliage or stem biomass.

One promising approach to quantifying spatially dependent fuel characteristics is to reduce point cloud complexity by collecting points into bins of gridded voxels. A voxel is a volumetric pixel that represents a value on a 3D grid [30]. Recent works have used voxelized point clouds to model tree canopies [31], estimate post-fire consumption and scorch [28,32], and to predict the distribution of mass and 3D structure of shrubs at high resolution [33].

In addition, voxelized point clouds are an important step towards 3D fuel cells that are used as inputs to coupled fire-atmosphere models. The wildland fuel cell concept was initially used to describe gridded fuel inputs in a computational domain [34,35]. The definition was expanded to describe patches of a fuel bed with distinct composition, characteristics, and architecture that become spatially independent beyond 0.5 m$^2$ [36]. More recently, the concept of a fuel cell has extended to three-dimensions in order to characterize the aggregation of interacting vegetation types [33].

Despite promising advancements in fire and fuels modeling, there are still significant gaps in research linking fuel models to the fire modeling environment. Transforming point clouds to three-dimensional fuel cells is a potentially direct link to gridded fuel inputs for coupled fire-atmosphere models. However, important questions remain about how to perform this transformation, as well as the scale to which modeled fire behavior is affected

by aggregated geometries—the combined spatial arrangement and integration of different fuel structures and shapes that collectively represent the fuel's physical configuration in the model.

Recent advances in remote sensing technologies, particularly Terrestrial Laser Scanning (TLS), offer sub-centimeter resolution in fuel descriptions. Such detailed descriptions are enticing for computational fire modelers due to their potential for improving model fidelity to observations. However, incorporating them directly into models requires new techniques, significant amounts of data, and large computational resources with no guarantee of better agreement with observed fire behavior.

This study is the first to evaluate how varying the resolution of LIDAR-based fuel descriptions impacts the agreement between modeled and observed fire behavior. Leveraging experimental heat treatments and high-resolution LIDAR scans collected at the Missoula Fire Sciences Laboratory, we systematically explore model sensitivity to fuel cell resolution when comparing FDS simulation results to observations of mass loss patterns in burning saplings. The numerical experiments are designed to reveal the lowest resolution required to match observations of fire behavior.

We find that fuel cell resolutions much coarser than the raw LIDAR point cloud (up to 8 cm) produce high fidelity model predictions of mass loss dynamics. This provides an efficient technique for reducing the complexity of high-resolution LIDAR data into computationally manageable inputs for 3D fire models like FDS. However, our results show that at the coarsest fuel cell resolutions examined (16 cm and above), achieving a close agreement between simulated and observed mass loss patterns requires non-physical adjustments to input parameters like bulk density. This indicates potential limitations to coarse fuel representations. Overall, this work enables integrating detailed LIDAR scans into 3D fire models, while revealing sufficient resolutions for matching simulated and observed fire behavior at laboratory scales.

## 2. Methods

### 2.1. Laboratory Experiments

The burn experiment reported in this paper was conducted in the Missoula Fire Sciences Laboratory burn chamber in July and August 2021. For the experiment, we acquired saplings of two species, Engelmann spruce (*Picea engelmannii*) and Ponderosa pine (*Pinus ponderosa*). Saplings were acquired in May of 2021 and stored in planter containers filled with soil, with roots intact. The trees were stored in a greenhouse from the time of their acquisition until their burn day. These burn experiments with live saplings were intended to collect data on both tree physiology and fire effects, and on fuel geometry and fire behavior. We report on the latter part of this study here.

Each day of the experiment, saplings were transported to the burn chamber and exposed to one of two heat treatments: low or high. The low heat treatment samples are defined as a flow rate of 10.8 liters of propane per minute for a heat release rate per unit area of 516.22 kW/m$^2$, and the high heat treatment samples are defined as a flow rate of 21.6 L of propane per minute for a heat release rate per unit area of 1032.46 kW/m$^2$. Both heat treatments lasted for 30 s. Figure 1 compares pictures of an example Engelmann spruce sapling before and after a high heat treatment. This example is a representative sample of the observed burn outcomes for many of the saplings measured in the experiment.

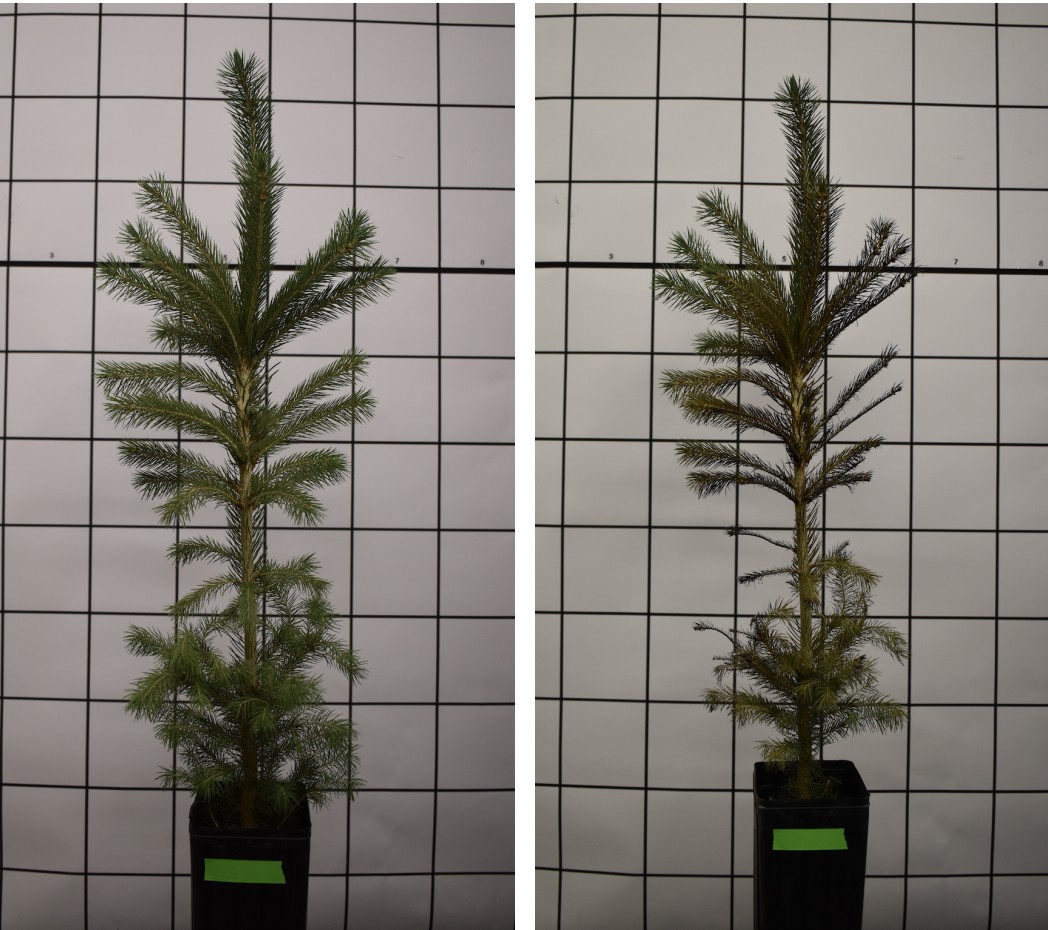

**Figure 1.** A side-by-side comparison of spruce sapling S63 before (**left**) and after (**right**) undergoing 30 s of high heat treatment. The images clearly illustrate the visual changes in the sapling's structure and foliage resulting from the heat treatment. The background grid shows squares of size 10 cm by 10 cm.

Within the burn chamber, a pair of concentric ring burners were placed over a piece of fibrous cement with a 12.7 cm diameter hole cut in the center. The inner burner ring had a radius of 14.6 cm, and the outer burner ring had a radius of 22.3 cm. Both burners had a diameter of 1 cm. At the time of the heat treatment, a sapling was placed through the opening such that it rested on a load balance. The height of the apparatus was adjusted so that the base of the planter container was level with the opening. This ensured a well-defined rule for the exposure of each sapling to the burners, but did not guarantee the same vertical distance between the bottom of the crown and the burners across saplings.

For each sapling, three-dimensional scans were collected from a Leica Geosystems BLK360 Terrestrial Laser Scanner, Heerbrugg, Switzerland. The TLS was run at a high density setting with a reported resolution of 5 mm. Two scans were taken from the same location on either side of a sapling before and after the burn treatment. The two scans were co-registered using Cyclone Register 360 BLK edition software version 2021.1.0 (Build r19986) from Leica Geosystems in order to create a single 3D point cloud. In addition, we recorded the weight of the sapling before, during, and after the burning period. The weight of the sapling during the burning period was measured with a load balance recording at approximately 0.5 Hz. Figure 2 shows a photograph of the experimental setup with labeled load balance, TLS device, and ring burner shortly before exposing the sapling to a heat treatment.

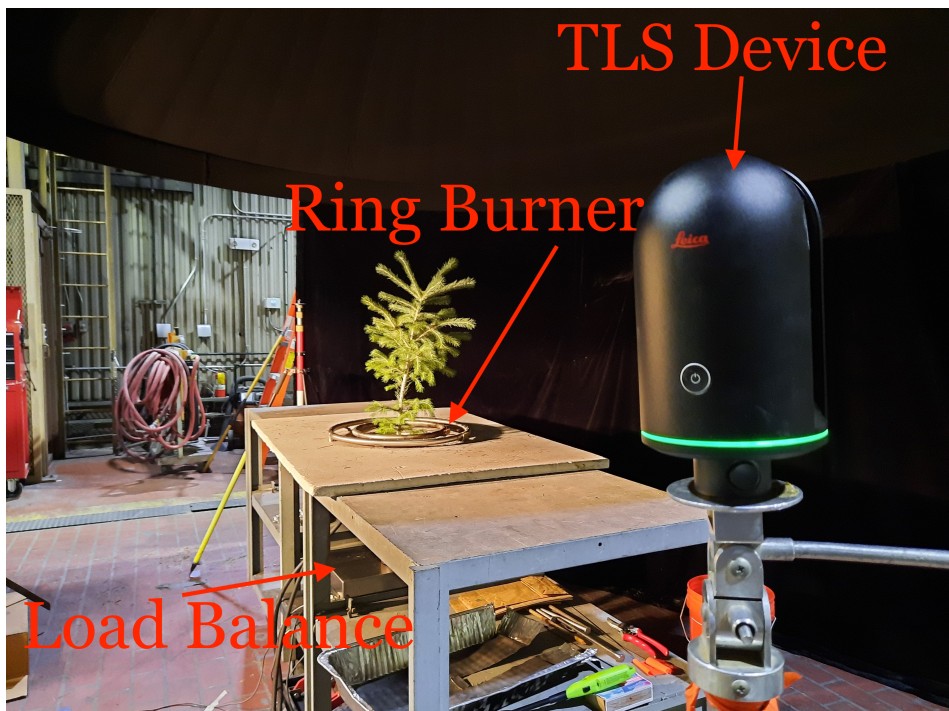

**Figure 2.** A labeled photograph of the experimental setup at the Missoula Fire Sciences Laboratory, illustrating the key components and their arrangement. The Terrestrial LIDAR Scanner is positioned to capture high-resolution point cloud data of the sapling, while the load balance measures the sapling's weight during the burning process. The ring burners provide controlled heat treatment, and an example sapling is shown in the testing area.

### 2.2. Converting Point Clouds to Fuel Cells

With the 3D point cloud data acquired from the Terrestrial Laser Scanner for each sapling, our objective was to transform this data into a format that can be effectively used in fire modeling applications. In particular, we need to transform the point cloud data into fuel cells, which are voxels containing vegetative attributes such as mass and moisture content.

In order to convert point cloud data to fuel cells, we began by applying a series of pre-processing steps to the point cloud data. The co-registered TLS point clouds contained points from the entire burn chamber, so we first reduced the 3D point cloud to a domain containing only point returns within the sapling extent. Outlier returns were present in the cropped point clouds due to residual soot from previous burns. To account for these erroneous points, we applied the Point Cloud Outlier Removal tool in the Open3D Python library [37] to filter points farther away from their neighbors compared to the average for the point cloud. Points were filtered more aggressively in the lower third of the sapling to account for a higher accumulation of soot near the burner surface.

After pre-processing, we applied a novel voxelization method to represent a collection of points as a single voxel. In general, voxelization involves the discretization of 3D space into a grid. The spatial resolution of the voxelized grid and values of grid cells are important considerations for working with point cloud data. For the purpose of converting point clouds to fuel cells appropriate for fire model inputs, we developed a technique of voxelizing at high spatial fidelity in order to capture geometries present in the LIDAR data, and then reducing to coarse fuel aggregations for practical use cases.

Our voxelization technique began by reducing the point cloud to its smallest possible voxel representation, given the physical constraints of the scanning device and environment. In this experiment we chose a 1 cm × 1 cm × 1 cm regular grid due to the reported 5 mm resolution of the TLS device. Higher resolution grids were attempted, but resulted in grid artifacts due to aliasing. We refer to an individual voxel in this grid as a reference voxel. Reference voxels were assigned a Boolean indicating the presence or absence of points

within the voxel. Thus, reference voxels containing multiple point returns were given the same positive value as a reference voxel containing one point return. This approach maintains the geometry of the sapling while avoiding issues associated with point density, such as variation in point density due to scanning device limitations, occlusions, or noise from the environment, ensuring a more accurate representation of the sapling's structure.

Reference voxels are useful in the sense that they provide the structure for aggregating a fine resolution point cloud into a coarse 3D grid. We constructed voxels at resolutions coarser than 1 cm by discretizing a 3D grid in the point cloud domain. Then, for each voxel in the coarse grid we assigned a value equal to the number of reference voxels occurring inside the coarse voxel. This results in a 3D grid which maintains the aggregated geometry of the original point cloud, and which has a relative density within each voxel. Spatial pooling by factors of 2 on the original 1 cm grid were considered in this study, though other spatial pooling factors resulting in non-integer grid cell values could also be used. Figure 3 shows an example sapling voxelized at resolutions from 2 cm to 16 cm following our method.

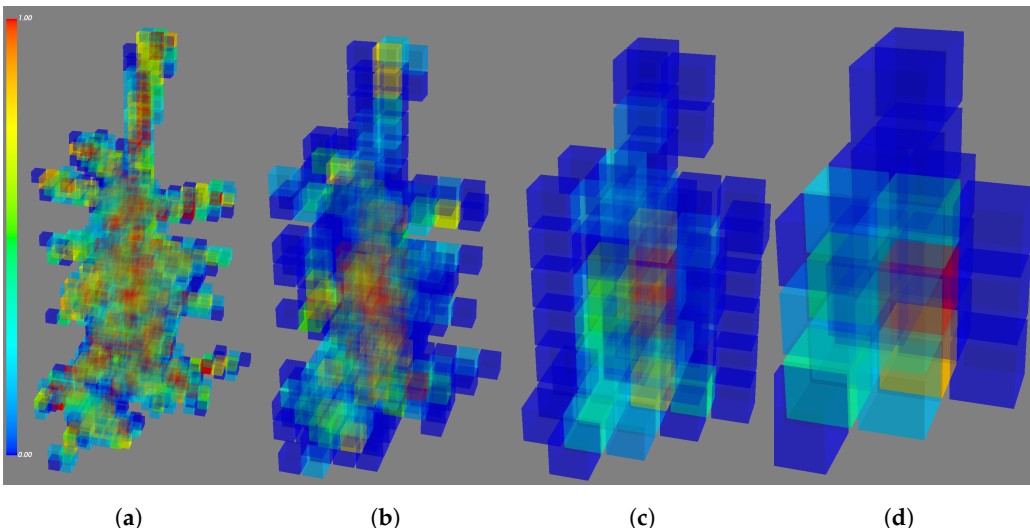

| (a) | (b) | (c) | (d) |

**Figure 3.** This figure provides a systematic visual comparison of point clouds processed through voxelization using the grid resolutions considered in this study. The subfigures, in a left-to-right sequence, exhibit grids with increasing coarseness: 2 cm, 4 cm, 8 cm, and 16 cm resolutions respectively. The accompanying colorbar, which scales from 0 to 1, serves as an indicator of the relative density within each voxel, quantifying the proportion of space occupied by reference voxels. A value of 0 represents an empty voxel, whereas a value of 1 denotes that the voxel is completely filled. (**a**) 2 cm Voxels. (**b**) 4 cm Voxels. (**c**) 8 cm Voxels. (**d**) 16 cm Voxels.

### 2.3. Fire Simulations

The goal of the voxelization technique described in Section 2.2 is to aggregate the spatial information present in high resolution point cloud data into fuel cells appropriate for use in coupled fire-atmosphere models. In this study we consider the application of point cloud derived fuel cells as fuel inputs to the Fire Dynamics Simulator (FDS) model.

FDS is a computational fluid dynamics model of fire-driven fluid flow. The model uses large-eddy simulation to numerically solve a form of the Navier-Stokes equations appropriate for low-speed, thermally driven flow, with an emphasis on smoke and heat transport from fires on a rectangular grid. In addition, FDS has an extensive validation suite [38] and has been used for similar comparisons between numerical simulation and burn chamber experiments [12].

FDS uses a particle model to represent objects that cannot be resolved on the numerical grid. In the particle model, different types and quantities of vegetation, like leaves, grass, and needles, are represented by a collection of subgrid particles that are heated via convection and radiation. This vegetation model assigns one weighted particle per grid cell

per fuel element to represent all of the actual particles of that vegetation class. The actual number of fuel elements, such as leaves or needles, represented by a particle is determined by the particle's mass per volume, or bulk density.

Based on observations from the experimental data, we assume that mass loss was driven by the drying and combustion of foliage and branch wood. To capture this process in the FDS model, we assign fuel cells representing both foliage and branch wood components using voxelized fuel grids as described in Section 2.2. This representation facilitates the modeling of vegetative biomass for both thermal degradation and heat transfer processes. In the model, it is sufficient to have one weighted particle per grid cell per fuel element to represent all of the actual particles of that vegetation class [39]. We place a particle representing foliage or branch wood in the FDS domain for each computational grid cell and each vegetative class present in a fuel cell.

In order to segment the LIDAR point cloud into the two vegetative class, we applied a threshold on the intensity value of each LIDAR point return [29]. Points with an intensity in the upper 60% of the threshold distribution were considered branch wood and points in the lower 40% of the threshold distribution were considered foliage. This threshold value was determined through visual inspection of the point clouds. We also ran experiments with percentage threshold ratios of 50\50 and 70\30 on a subset of the data without noticeable changes in the results. The segmented points were then processed into voxelized grids following the method presented in Section 2.2.

For each occupied voxel in the computational domain we provided the model with fuel attributes such as bulk density, fuel moisture content, particle density, and surface area to volume ratio. Bulk density, the quantity of fuel per unit volume, was computed for the foliage and branch wood components for each cell. To compute the bulk density of a component in a fuel cell, we took the total dry mass of the component for the entire sapling, divided the mass by the number of reference voxels in the sapling, and then multiplied the total dry mass per reference voxel by the number of reference voxels present in the fuel cell. This process is expressed by the equation:

$$\rho_{f,i} = \frac{m_c n_i}{N V_i} \tag{1}$$

where $\rho_{f,i}$ is the bulk density of fuel cell $i$ (kg/m$^3$), $m_c$ is the total dry mass of component $c$ (kg), $V_i$ is the volume of the fuel cell $i$ (m$^3$), $N$ is the dimensionless quantity of reference voxels present in the entire domain, and $n_i$ is the dimensionless quantity of reference voxels present in fuel cell $i$.

In this study, the total dry mass ($m_c$) of the sapling is conserved across different fuel cell resolutions. However, the bulk density ($\rho_{f,i}$) of a fuel cell varies as a function of fuel cell resolution. This variability arises due to the change in volume ($V_i$) that each fuel cell represents. As the volume of a fuel cell expands or contracts, the bulk density is adjusted to reflect a new fuel load per unit volume in order to maintain the conservation of mass of the sapling. It is noteworthy that this variability could have a significant impact on simulated fire behavior through a change in packing ratio, or the portion of the fuel cell volume that is occupied by fuel. Changes to the packing ratio of a fuel cell could impact simulated fire behavior by affecting heat transfer mechanisms and airflow patterns within the fuel cell [40].

Fuel moisture content was assumed to be constant across fuel cells and components for a given simulation. Section 2.4 details the estimation of component dry mass and fuel moisture content. All other fuel-related parameters are based on the NIST Douglas Fir experiments in the FDS validation guide version 6.7.9 [38].

All simulations were run with FDS version 6.7.9 released on 30 June 2022. Heat treatments lasted for 30 s of simulation time and modeled the heat release rate per unit area measured in the laboratory experiments. The boundary conditions were open on the sides and top of the domain, and were closed at the bottom of the domain. We modeled the simulation environment on a 0.6 m × 0.6 m × 1.2 m regular rectangular

grid with a cell resolution of $2 \, \text{cm} \times 2 \, \text{cm} \times 2 \, \text{cm}$. We note that the resolution of the computational grid in FDS is independent of fuel cell resolution. Fuel cells can have the same, or coarser, resolution as the computational grid. The total mass of the component fuel elements were output by FDS during the 30 s of simulated heat treatment at 100 Hz to allow for comparison to observations (FDS input file templates can be found in the following repository: https://github.com/amarcozzi/Application_LIDAR_Derived_Fuel_Cells (accessed on 9 October 2023)).

*2.4. Numerical Experiments*

To assess the uncertainty associated with fuel inputs to fire models, we conducted numerical experiments comparing simulated mass over time in FDS to mass over time data collected in the laboratory. First, we selected 16 saplings from the two species, 8 ponderosa and 8 spruce, and two heat treatments, 8 high and 8 low, for voxelization and analysis in our numerical experiments. Next, we used DAKOTA, a widely utilized software tool for optimization and uncertainty quantification in computational models, to perform a multidimensional parameter study [41]. This study was conducted through a grid search, a methodical approach where the parameter space is explored systematically by evaluating the model at a grid of points defined by the chosen variables. In our case, each point in the parameter space had three dimensions: dry foliage mass, fuel moisture content, and fuel cell resolution. The grid search approach allowed for systematic exploration across these variables, capturing their interdependencies and providing a detailed understanding of how they influence the model's outcomes.

The dry foliage mass was uniformly sampled at 16 points over the range $[5, 80]$g. An initial coarse grid search was conducted to determine the appropriate range of dry foliage mass values. Based on this, a range of $[5, 80]$g was selected so that a well-defined region of minima would be present across all model runs for a given fuel cell resolution and sapling type. This consistent range ensured that comparisons between simulation sets were meaningful for analyzing the effects of varying fuel cell resolution.

Due to the construction of the experiments we did not have reliable estimates of fuel moisture content for the saplings exposed to the heat treatments. As such, we estimated fuel moisture content by uniformly sampling 16 points over the range $[20, 150]$% according to estimates published in the Fire Behavior Field Reference Guide [42]. Fuel cell resolution was sampled from the set $\{2, 4, 8, 16\}$, where each resolution in the set is the length of the side of a cube, or voxel, in centimeters.

To further investigate the significance of the saplings' branching structure and void space, we additionally modeled the saplings as homogeneous cylinders in which fuel cells were distributed uniformly within a cylinder of a height and radius equal to that of the sapling. The height and radius of the cylinder were respectively computed as the distance between the vertical and horizontal extents of the processed point cloud.

For the uniform cylinder representation, a distinct and higher range for dry foliage mass was implemented, differing from the $[5, 80]$g range used for the resolutions in $\{2, 4, 8, 16\}$. To determine the range of dry foliage mass for each sapling we selected a mass range that resulted in a well-defined region of minima based on the initial coarse grid search. The adjustment to the mass range for the uniform cylinder was required due to the $[5, 80]$g range's inability to provide sufficient mass for convergence to a region of minima, as shown in Section 3.

For each of the sixteen saplings we sampled 1024 uniform three-dimensional points in the parameter space. An FDS simulation was run for every point, and the simulated mass over time was compared with the observed mass for the corresponding sapling and heat treatment. This process resulted in 1024 observations of fit between the simulated and observed data for each of the sixteen saplings.

In order to compare the results from our parameter sweep with the observed data we calculated the Root Mean Squared Error (RMSE) for each simulation using the mass over time data generated by the FDS model and the mass over time data collected in

the laboratory. This provided a quantifiable measure of the goodness of fit between the simulated and observed mass over time data, offering insights into the performance of the FDS model when simulating mass loss dynamics. The RMSE for each pair of model output and observed data was computed as

$$\text{RMSE} = \sqrt{\frac{1}{N} \sum_{i=1}^{N} (m_o^i - m_m^i)^2}. \tag{2}$$

where $m_o^i$ is the observed mass from the experimental heat treatment at time step $i$, and $m_m^i$ is the mass in the FDS simulation at time step $i$. The RMSE was computed using 30 s intervals corresponding to the duration of the laboratory heat treatment and the FDS simulation.

To validate the alignment between simulated and observed data, and to estimate additional model parameters, we leveraged information collected from destructively measured saplings left over from the heat treatments. Ten ponderosa and fourteen spruce saplings were deconstructed into foliage and branch components, oven dried, and then weighed. Additional data such as basal diameter, height, and branch count were collected for both the deconstructed saplings and the saplings exposed to the heat treatment. We fit two linear models using least squares to predict dry foliage mass and dry branch mass using basal diameter and height as independent variables. The total dry branch wood mass was estimated with the linear model and distributed to fuel cells using the reference voxel approach as previously described. For each sapling, we estimated dry foliage mass with the linear model and compared the estimate with the results of the parameter sweep.

Throughout the experiment the load balance was actively logging data before, during, and after the heat treatment. However, load balance logging was independent of heat treatment start and stop times. Additionally, there were different procedures for exposing the spruce and ponderosa saplings to the heat treatment. Spruce saplings were lowered onto the load balance after the ring ignitors had reached an equilibrium burn rate, while ponderosa saplings were resting on the load balance at the beginning of the heat treatment.

Given the variations in the experimental setup, especially the independent logging of load balance and differences in heat treatment procedures, it was necessary to derive rules to constrain the mass over time data to a consistent thirty-second heat treatment period. For the spruce saplings, we took the maximum mass, corresponding to the equilibrium reached just after the sapling made contact with the load balance, and set the corresponding index to time zero. For the ponderosa saplings, we took the minimum mass, occurring before a slight rise in mass due to the end of an upward buoyant force at the end of the heat treatment, and set the corresponding index to time thirty seconds. This process resulted in consistent thirty second mass over time intervals for each heat treatment. However, the resulting data likely does not capture the full extent of the burn period, as lab technicians manually closed the valve of the propane tank at the 30 s mark, leading to slightly inconsistent heat treatment lengths on the order of one to three seconds.

Based on the numerical experiments and resulting measurements of goodness of fit, the study examined three key aspects:

1. The alignment between physically plausible input parameters and model outputs, assessing how well the latter fit the observations.
2. The effectiveness of the voxelization methodology used in employing TLS data to inform fire model inputs.
3. The sensitivity of the results to the chosen input parameters, testing how variations in these parameters influenced the outcomes.

## 3. Results

The FDS model is able to reproduce observed mass loss patterns from TLS-informed inputs in terms of curve shape and mass loss quantities. Figure 4a shows the best agreement within the 2–16 cm fuel cell range between simulated and observed data using our

voxelization technique. This simulation used 16 cm fuel cells from spruce sapling S63 with a dry foliage mass of 14.8 g and a fuel moisture content of 92 %. The RMSE from this simulation was 0.75 g. A close agreement between the simulated and observed mass loss is observed in terms of the curve shape and the resulting mass change, highlighting the FDS model's capability to accurately reproduce observed mass loss patterns by utilizing TLS data to inform fuel inputs.

Figure 4b shows the highest minimum RMSE value in Table 1. This simulation used 2 cm fuel cells from sapling S16, and had an RMSE of 5.04 g. While this simulation resulted in the highest minimum RMSE in our parameter sweep, we continue to observe a close match between the simulated and observed mass over time curves.

The parameter sweep revealed a wide range of RMSE values across the different saplings, as shown in Table 1. These variations in minimum RMSE values for individual saplings highlight that the goodness of fit is influenced by factors specific to each specimen, such as branch geometry, physiology, and the precision of laboratory measurements like the load balance and TLS scanner.

Furthermore, Table 1 demonstrates that RMSE values are relatively consistent across all considered fuel cell resolutions, indicating that the model's ability to closely match the observed data is not heavily influenced by changes in fuel cell size. The maximum variation for ponderosa and spruce sapling RMSE for all considered fuel cells was 1.06 g for spruce saplings and 1.95 g for ponderosa saplings. These values reflect the highest quantity of spread in the RMSE values across fuel cell resolutions within the respective groups of saplings and underscore the stability of the model's approximation of observed mass loss patterns across different resolutions.

**Table 1.** Minimum RMSE values in grams for different fuel cell resolutions and the uniform cylindrical representation across 16 saplings, including 8 ponderosa (P) and 8 spruce (S) saplings. The table shows mean values for each resolution and sapling, as well as separate RMSE means for the ponderosa and spruce sapling groups.

| Tree | 2 cm | 4 cm | 8 cm | 16 cm | Cylinder | Mean |
|------|------|------|------|-------|----------|------|
| P06 | 1.55 | 1.41 | 1.31 | 1.23 | 1.05 | 1.31 |
| P07 | 1.34 | 1.48 | 1.99 | 2.13 | 1.18 | 1.62 |
| P11 | 2.13 | 2.04 | 2.00 | 1.71 | 1.56 | 1.89 |
| P14 | 3.18 | 2.93 | 2.78 | 2.20 | 1.23 | 2.46 |
| P15 | 4.48 | 4.70 | 4.76 | 4.73 | 5.92 | 4.92 |
| P17 | 5.01 | 5.00 | 4.74 | 3.85 | 3.61 | 4.44 |
| P31 | 1.19 | 1.18 | 1.38 | 1.19 | 1.41 | 1.27 |
| P36 | 3.04 | 3.09 | 2.83 | 3.24 | 2.99 | 3.04 |
| **P Mean** | 2.74 | 2.73 | 2.72 | 2.55 | 2.37 | 2.62 |
| S16 | 5.04 | 4.74 | 4.66 | 4.33 | 4.83 | 4.72 |
| S30 | 3.97 | 3.82 | 3.59 | 3.04 | 2.91 | 3.47 |
| S31 | 3.25 | 2.96 | 2.68 | 2.76 | 3.24 | 2.98 |
| S48 | 3.75 | 3.86 | 4.08 | 3.85 | 4.22 | 3.95 |
| S50 | 4.14 | 3.99 | 4.12 | 3.69 | 3.82 | 3.95 |
| S51 | 2.58 | 2.68 | 2.53 | 2.37 | 2.43 | 2.52 |
| S58 | 4.00 | 3.89 | 3.80 | 4.16 | 4.25 | 4.02 |
| S63 | 1.01 | 1.08 | 1.44 | 0.75 | 0.53 | 0.96 |
| **S Mean** | 3.47 | 3.37 | 3.36 | 3.12 | 3.28 | 3.32 |

The 16 cm fuel cell resolution and the uniform cylinder representation generally yielded the lowest RMSE on average. This observation suggests that these representations might offer a more accurate approximation of a sapling's mass loss behavior in the FDS model. However, the table does not explore the relationships between dry foliage mass, fuel moisture content, and minimum RMSE for each sapling and fuel cell resolution. To thoroughly evaluate the model's fit to observational data and parameter sensitivity,

we further analyze the relationships between dry foliage mass, fuel moisture content, and minimum RMSE for each sapling and fuel cell resolution.

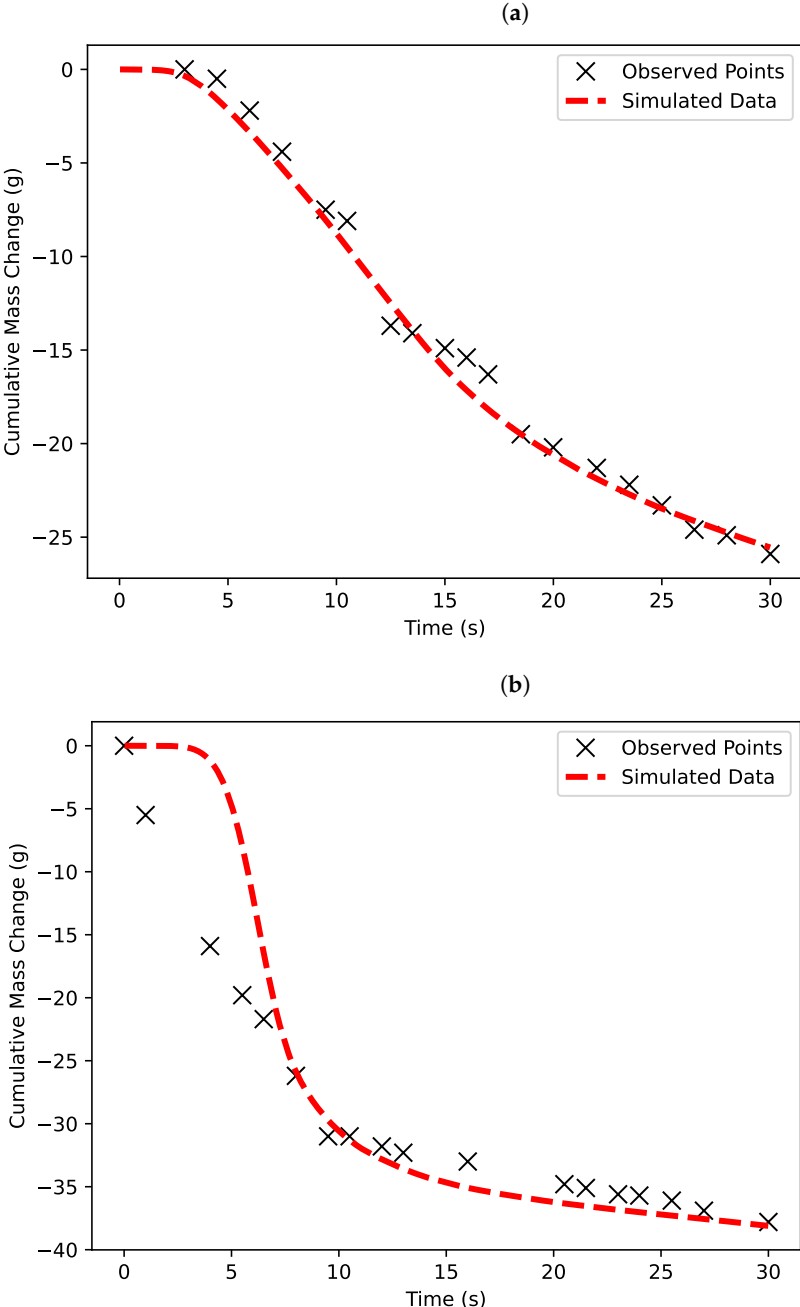

**Figure 4.** Cumulative mass change comparisons for Saplings S63 and S16, with both simulated and observed data plotted on the same graphs. The red dashed line represents the FDS fire model's predictions, while the black x marks corresponds to the experimental measurements obtained using a load balance. The comparisons showcase the best and worst fits among the minimum RMSE model runs featured in Table 1, not including the uniform cylinder column, indicating the range of agreement between the simulated and observed data. (**a**) Sapling S63 with 16 cm fuel cells, representing the lowest minimum RMSE in the 2 cm to 16 cm range in Table 1. (**b**) Sapling S16 with 2 cm fuel cells, representing the highest minimum RMSE in Table 1.

We consider the relationships between dry foliage mass, fuel moisture content, and minimum RMSE for sapling P17 in Figure 5. In this case, a well-defined region of consistent RMSE minima is evident across all fuel cell resolutions. The model shows high sensitivity to dry foliage mass inputs with regions of high RMSE on both the lower and upper ends of the range considered. Additionally, the model is sensitive to fuel moisture content, though the effect is lower than the sensitivity to foliage mass and increases at high values of dry foliage mass.

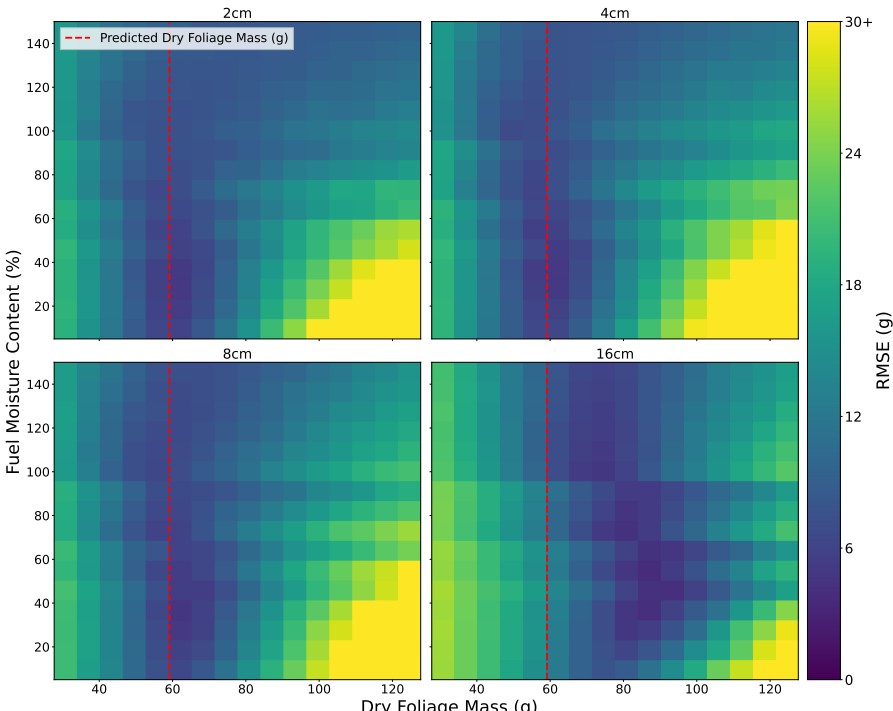

**Figure 5.** Distribution of RMSE values across the full parameter sweep of 1024 samples for sapling P17. Each pane represents simulations with different fuel cell resolutions: upper left (2 cm), upper right (4 cm), bottom left (8 cm), and bottom right (16 cm). The x and y axes of each pane correspond to the sampled range of dry foliage mass and fuel moisture content, respectively. Pixel colors indicate the RMSE values resulting from the comparison between the simulated model output and the observed data for sapling P17, with darker colors representing lower RMSE values. The dashed red line represents the predicted dry foliage mass from the linear model trained on deconstructed saplings.

Although a region of minimum RMSE is also evident at 16 cm resolution, the model fit is relatively poor and the dry foliage mass input required to optimize fit to observational data is much higher than for the 2, 4, and 8 cm resolutions. Furthermore, the additional mass needed to enhance the model fit to experimental data diverges from the best-fit estimates of dry foliage mass predicted by the linear model trained on deconstructed saplings. The 2, 4, and 8 cm resolution fuel cells are on average 1.88 g away from the predicted dry foliage mass, whereas the 16 cm fuel cell minimum is 28.55 g away from the predicted dry foliage mass.

These observations suggest that modeled fire behavior diverges at fuel cell resolutions between 8 cm and 16 cm. For coarser fuel cell resolutions optimizing the fit between modeled and observed data comes at the cost of diminishing the likelihood that the physical parameters input to the model accurately represent the actual dry foliage mass of a sapling.

When these findings are extended to all saplings via a Kernel Density Estimate, using the dry foliage mass corresponding to the minimum RMSE for each fuel cell resolution, as shown in Figure 6, it becomes evident that dry foliage mass at minimum RMSE for 2, 4, and 8 cm fuel cells follow similar distributions. The peaks of the Kernel Density Estimate for these resolutions fall within a narrow range (47.13 g for 2 cm, 45.69 g for 4 cm,

and 45.34 g for 8 cm), suggesting that optimal foliage mass at the 2 cm resolution is likely to be similar to optimal mass at the 4 cm and 8 cm resolutions.

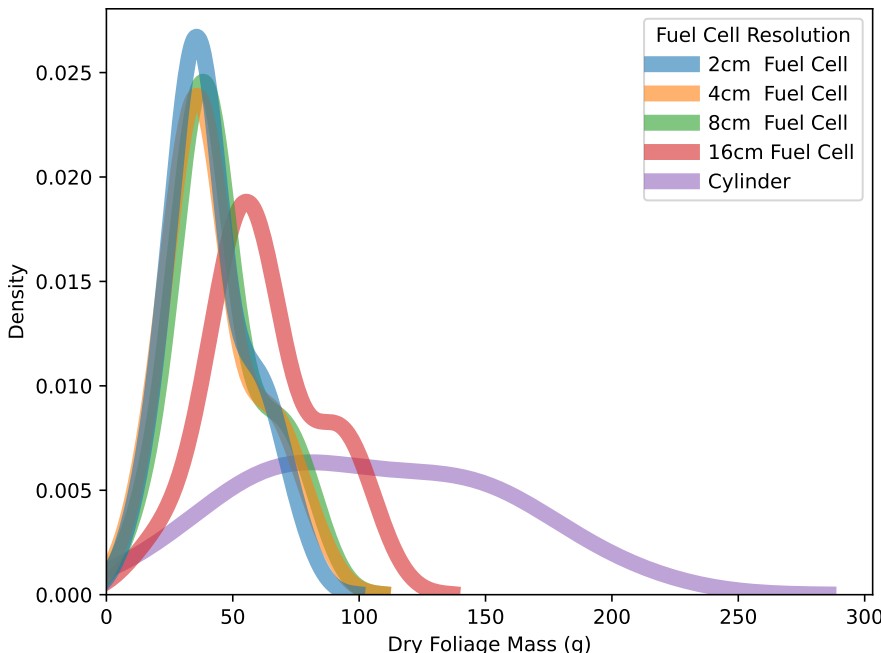

**Figure 6.** Kernel density estimate (KDE) plots of the dry foliage mass associated with the minimum RMSE for each fuel cell resolution in the total set of 16 saplings. Each curve represents the distribution of dry foliage mass at the minimum RMSE for a specific fuel cell resolution, highlighting the consistency in mass values across 2, 4, and 8 cm resolutions and the deviation observed at the 16 cm resolution and uniform cylinder geometry.

Between the 8 cm and 16 cm fuel cell resolutions, there is a marked shift rightward in the mass required to achieve a suitable fit between simulated fire behavior and observed mass data, reflected in a higher mean dry foliage mass of 58.05 g for the 16 cm fuel cell resolution. Notably, the peak of the uniform cylinder distribution for the uniform cylinder is at 104 g of dry foliage mass, more than double the mass at the peaks for the 2–8 cm fuel cell resolutions.

Furthermore, the range of optimal dry foliage mass expands as resolution coarsens, indicating a less consistent relationship between mass and minimum RMSE at coarser resolutions. This can be seen in Figure 6 as a decrease in the sharpness of peaks at 16cm resolution and in the uniform cylinder representations.

Finally, FDS model performance was evaluated across fuel cell resolutions and saplings by comparing predicted dry foliage mass with minimum RMSE for each sapling and fuel cell resolution in Figure 7. For spruce saplings in Figure 7a the mean absolute distances vary across resolutions: 2 cm fuel cell at 11.95 g, 4 cm at 12.54 g, 8 cm at 14.91 g, 16 cm at 29.05 g, and the uniform cylinder at 53.86 g. In comparison, ponderosa saplings in Figure 7b show close agreement for 2 cm, 4 cm, and 8 cm fuel cells with mean absolute distances of 6.34 g, 3.84 g, and 2.17 g respectively. However, model performance decreases at 16 cm resolution and the uniform cylinder representation with mean absolute distances of 18.66 g and 81.99 g.

The observed data show a consistent trend: as fuel cell resolution coarsens, the mean absolute distance between the predicted dry foliage mass and the mass at minimum RMSE increases. For ponderosa saplings, the increase is more pronounced at coarser resolutions, particularly in the uniform cylinder representation. Spruce saplings exhibit larger distances across all resolutions, though we still observe the underlying phenomenon of increasing mean absolute difference from the predicted dry foliage mass as fuel cell resolution coarsens. This trend highlights an inherent challenge in the model, as coarser resolutions require

increasingly implausible input parameters to minimize RMSE. This behavior, consistent across both species, calls for for further investigation into the effects of resolution on model fidelity.

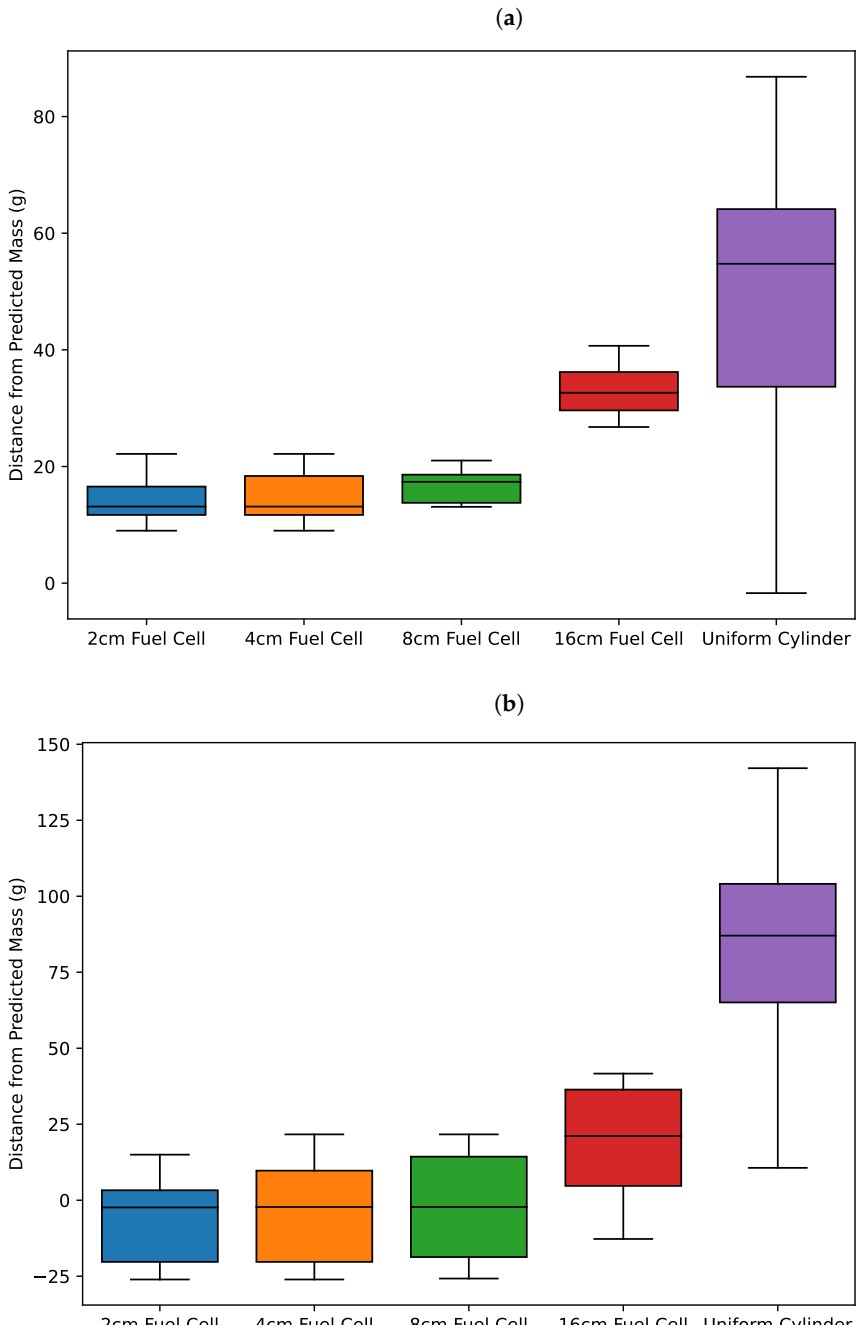

**Figure 7.** Distance from the predicted dry foliage mass to the minimum RMSE dry foliage mass for each fuel sapling and fuel cell resolution. The y-axis shows the difference in grams between the minimum RMSE dry foliage mass and the predicted dry foliage mass from the linear model. Each box corresponds to a fuel cell resolution aggregated across saplings of the same species. Boxes extend from the first quartile to the third quartile of the data, with a horizontal line at the median. Whiskers extend from the box by 1.5x the inter-quartile range. (**a**) Spruce Saplings. (**b**) Ponderosa Saplings.

## 4. Discussion

This study aimed to evaluate the efficacy of the Fire Dynamics Simulator (FDS) model in simulating the dynamics of mass loss in ponderosa and spruce saplings. The model

was informed by Terrestrial Laser Scanning (TLS) data, which was used to guide fuel descriptions. Our work provides a method for reducing the large volumes of data in TLS point clouds to fuel descriptions that have fidelity to both the geometric complexity and composition of the modeled saplings. The result is well suited for input into fire behavior models such as FDS. We find that the FDS model can accurately replicate observed mass loss patterns when provided with TLS-derived fuel data via our voxelization method. This outcome highlights the potential of our approach in integrating detailed TLS data into a 3D fire model like FDS, thereby potentially improving the precision of vegetation representation in fire simulations. However, our results suggest that the fuel model includes enough internal degrees of freedom that it is possible to closely match observational data using non-physical input parameters.

Our results also revealed a complex relationship between fuel cell resolution, dry foliage mass, and the accuracy of the FDS model in replicating observed mass loss dynamics. While we achieved a close correlation between simulated and observed mass loss across various scales of fuel cell resolution, a trade-off emerged between model accuracy and the fidelity of input parameters to physically meaningful input parameters as fuel cell resolution coarsened. This finding raises questions about the appropriateness of assumptions of fuel homogeneity when representing vegetation in fire behavior models at fine scales.

These results are consistent with previous work [16] that emphasize the significance of spatial heterogeneity in determining fire behavior. Our results suggest that using simple geometries like cones, frustums, or cylinders to represent complex vegetation may not be appropriate at finer scales. This conclusion is particularly relevant given the increasing availability of high-resolution LIDAR data, which can capture vegetation structure more accurately than simpler geometric representations, and an increased interest in applying couple fire-atmosphere models to practical fire and forest management problems.

Our research not only corroborates the significance of spatial heterogeneity in modeled fire behavior, but also explores the varying scales at which it influences the agreement between observed and modeled mass loss. While our study results are applicable to fuel cells and underlying grid resolutions suitable for small laboratory-scale domains, many practical applications of 3D coupled-atmosphere fire models focus on scales more appropriate for modeling prescribed fires and wildfire events. In these larger scale scenarios, the range of fuel cell resolutions considered in this study becomes computationally prohibitive for large domains. Therefore, it may be beneficial to investigate how the relationships observed at the laboratory scale between fuel cell resolution, underlying grid resolution, and fire behavior could inform decisions in field-scale applications.

We observed a close correlation between simulated fire behavior and observed mass loss for fuel cell resolutions of 2 cm, 4 cm, and 8 cm when simulated on a regular grid with a 2 cm resolution. This finding suggests that ratios of 1:1, 2:1, and 3:1 between fuel cell resolution and underlying grid resolution may be suitable for accurately modeling fire behavior in these laboratory-scale scenarios. However, our results also indicate that ratios above 3:1 led to significant changes in the modeled fire behavior. For field-scale applications, fire modelers might maintain a similar ratio between fuel cell resolution and underlying grid resolution to achieve a sufficient representation of complex vegetative geometry. For instance, in a practical scenario with a 1-m grid resolution, a fire modeler might select a fuel cell resolution of 1 m 2 m, or 3 m, informed by the laboratory-scale findings of this study. This approach could help balance computational efficiency, data assimilation, and accurate representation of vegetative geometry in the model.

However, it is important to recognize that applying the observed relationships from laboratory-scale scenarios to field-scale applications may not be universally applicable. Therefore, further research is needed to explore how spatial heterogeneity scales in larger domains and how the optimal ratio between fuel cell resolution and underlying grid resolution might vary across different contexts. Investigating these factors could ultimately contribute to more informed decision-making in fire modeling and improve the accuracy and reliability of fire behavior predictions in diverse settings.

In addition, we report significant differences in the mean RMSE between spruce and ponderosa sapling categories. A possible reason for this phenomena is that structural differences between the two species result in significantly different fuel cell geometry and continuity in our three-dimensional models. In this case the long needles of the ponderosa saplings led to relatively continuous fuel models compared to the spruce fuel models with more void space. While not further explored in this study, another promising avenue for future research is to apply recent developments in point cloud processing to relate crown continuity to fire behavior in both modeled and real-world environments [43,44].

While these future research directions hold promise, it is equally crucial to acknowledge the limitations of our current study that may have influenced our findings. Firstly, our study did not account for the buoyant forces that arise due to vertical air flows around the sapling during combustion. Buoyant forces, a result of the density difference between the heated air surrounding the burning sapling and the cooler ambient air, can significantly influence the dynamics of fire behavior. Specifically, these forces can induce upward movement of air and flame, potentially leading to an observed decrease in mass during the heat treatment due to the entrainment of cooler air. The exclusion of this complex, dynamic phenomenon from our study may have implications for the interpretation of our results, and future research should consider its inclusion to enhance the realism and accuracy of fire behavior modeling.

Secondly, our voxelization method exhibited a tendency to oversample the stem of the tree. This oversampling issue is a common challenge when using LIDAR data for vegetation structure representation. LIDAR data, due to its high spatial resolution, often captures more detail of the stem compared to the foliage, leading to an over-representation of the stem in the derived 3D model. This occurred in our study despite the implementation of our foliage and branch wood segmentation process, which had a negligible impact on the simulation results.

Although the overall proportions of modeled foliage consumption were consistent with laboratory data, we observed increased consumption of foliage and branch wood along the central axis of the tree stem compared to what was observed experimentally. This discrepancy in heat release rate and stem consumption due to oversampling the stem of the tree could potentially influence the accuracy of fire behavior predictions. Future research endeavors could focus on the development and application of more refined segmentation processes or alternative methods to balance the representation of different tree components. This could include techniques for better distinguishing between stem and foliage in LIDAR data or adjusting the voxelization process to account for the oversampling of the stem.

It is also possible that the rules defining the period of heat treatment for observed mass loss data may have contributed to systematic errors between simulated and observed results. Some heat treatment periods used in our analysis may not align precisely with the laboratory data due to inconsistencies in how the saplings were placed on the load balance and the time it took to manually turn of the burner. These potential discrepancies might explain the differences observed between Figure 4a,b, where a higher RMSE in Figure 4b corresponds to an immediate drop in observed mass and a delayed drop in simulated mass.

Despite these limitations, we still find close agreement between our observed and simulated data, and are encouraged by the potential application of our voxelization technique for making a direct connection between LIDAR data and fuel input to 3D fire models. These limitations not only provide context for the interpretation of our findings but also highlight potential avenues for future research.

## 5. Conclusions and Summary

This study introduces a new method in fuel and fire behavior modeling, focusing on the dynamics of mass loss in ponderosa and spruce saplings through the use of the Fire Dynamics Simulator (FDS) model. Our approach is the first step towards integrating high resolution Terrestrial Laser Scanning (TLS) data into detailed fuel descriptions for coupled fire-atmosphere models. We identify and present a novel voxelization technique



for effectively reducing the TLS data volume, making it manageable for integration into the FDS model, and potentially improving the accuracy of vegetation representation in fire simulations.

The results showed that the FDS model can replicate observed mass loss patterns with accuracy when provided with TLS-derived fuel data through the voxelization method. However, a trade-off was observed: as the fuel cell resolution coarsened, the model maintained accuracy, but this required the use of increasingly non-physical input parameters. This indicates that lower geometric fidelity is achievable, but comes at the cost of the realism of input parameters. This finding prompts questions regarding the assumptions of fuel homogeneity when representing vegetation at fine scales and underlines the importance of spatial heterogeneity in fire behavior modeling. Additionally, the study suggests that maintaining an optimal ratio between fuel cell resolution and underlying grid resolution may be beneficial for balancing computational efficiency, data assimilation, and accurate representation of vegetative geometry in both laboratory and potential field applications.

**Author Contributions:** Conceptualization, A.A.M., J.V.J. and R.A.P.; methodology, A.A.M., J.V.J., C.A.S. and J.Z.D.; software, A.A.M. and J.Z.D.; validation, A.A.M.; formal analysis, A.A.M.; investigation, S.J.F., R.A.P. and A.A.M.; resources, S.J.F.; data curation, S.J.F. and A.A.M.; writing—original draft preparation, A.A.M.; writing—review and editing, A.A.M.; visualization, A.A.M. and J.Z.D.; supervision, J.V.J. and C.A.S.; project administration, J.V.J. and R.A.P.; funding acquisition, R.A.P. All authors have read and agreed to the published version of the manuscript.

**Funding:** This research was funded by National Science Foundation EPSCoR Cooperative Agreement OIA-2119689, and the Strategic Environmental Research and Development Program (SERDP) Closing Gaps Project (RC20-1025) and 3DFuels Project (RC19-1064). We gratefully acknowledge their support for this work.

**Data Availability Statement:** The data presented in this study are available on request from the corresponding author. Template input files and analysis code is publicly available at: https://github.com/amarcozzi/Application_LIDAR_Derived_Fuel_Cells (accessed on 9 October 2023).

**Acknowledgments:** The authors thank the staff and laboratory technicians at the Missoula Fire Sciences laboratory for their hard work and dedication to the experiment.

**Conflicts of Interest:** The authors declare no conflict of interest. The funders had no role in the design of the study; in the collection, analyses, or interpretation of data; in the writing of the manuscript, or in the decision to publish the results.

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
