# Peer review of "Application of LiDAR Derived Fuel Cells to Wildfire Modeling at Laboratory Scale"

_fire, doi:10.3390/fire6100394_

Round 1
Reviewer 1 Report
This manuscript describes laboratory experiments in which saplings of ponderosa and spruce (eight of each) were burned after the spatial location of foliage and roundwood was derived via terrestrial LiDAR scans. Two different ignition procedures were used (called high and low). Three-dimensional, time dependent, numerical simulations of the sapling burns were conducted using the Fire Dynamics Simulator. For most of the simulations, the location of vegetation was based on LiDAR. Some simulations assumed the sapling had a uniform cylindrical shape. Comparison of simulation results to observations was based solely on the root mean squared error measure of the difference between a 30 second time history of the simulated and observed mass.
For each sapling case in which the vegetation location was derived from LiDAR, 1024 simulations were conducted. This large number of simulations was the result of a simulation for 16 values of dry foliage mass, 16 values of moisture, and 4 values of spatial resolution for the vegetation (16x16x4=1024).
The procedures and results are of interest. However, the manuscript needs improvement. This includes clarity and fullness in reporting the methodology, choices made, and the generality of the implications of the results. These are significant enough to warrant major revision. Hopefully, the comments, questions, and suggestions I’ve made are helpful.
Comments and questions have been provided on a marked-up version of the manuscript pdf. There are additional comments below, some are repeated in the marked-up pdf file.
1. Please include units. This is fundamental!
2. Overall, this work raises the question of why a much smaller set of simulations (compared to 1024) was not conducted, for each sapling, that focused more closely around the measured value of moisture and measurement-based estimates of the total foliage and roundwood in the sapling. The same range of fuel cell sizes could have been used.
3. I may have misunderstood what was done, but it appears that as the fuel cell resolution decreased it was ensured that the total mass in the simulated sapling would remain constant (as it should), but it’s not clear that the bulk density did or could. If this is the case, it needs to be stated and the impact of this on the simulated fire behavior discussed.
4. It appears that the RMSE was computed for the duration of the burner (30 s). If this is the case and it’s not already stated explicitly, it needs to be stated. Also, if this is the case, why? Presumably, consumption continued past the 30 s (at least for some cases, as is implied in Fig. 4a). A more complete testing of the model would include times in which heat exposure was no driven by the ignitor.
5. There seems to be very little discussion about the dependence of the results (numerical or experimental) on the high vs. low ignition procedure. Were the results insensitive to ignition procedure?
6. The scale on Figure 8 makes it difficult to see how relevant moisture was to mass consumption (for the 2, 4, 6 cm fuel cells) along the assumed best estimate of total dry foliage mass (the vertical dashed red line; also, it would be helpful to have a horizontal line showing the moisture value for this sapling). Please elaborate. If the simulation results are not very sensitive to moisture, why?

Author Response
We would like to express our sincere gratitude to Reviewer #1 for their time and effort to deeply read, think about, and provide feedback on our manuscript. Reviewer #1 brought up six major points that we can address to better clarify the scope and goals of our work, as well as many comments to help us address specific lines or sections of the manuscript that can be improved. We are grateful for both forms of reviewer feedback and we believe that their comments will strengthen our research conclusions and the ability to effectively communicate our findings.
Please see the attached .pdf file for our response to the reviewer's comments.

Reviewer 2 Report
Please refer to the attached comments.

Round 2
Reviewer 2 Report
Refer to the attached file.

Author Response
Thank you again for the feedback to our manuscript. We have addressed the feedback related to the abstract by modifying the manuscript to be more specific about which resolutions were insensitive to model results, and which resolution we noticed a deviation in model behavior.